# Influence of main parameters on the displacement process by spontaneous imbibition based on LBM

Tianju Wang[1], Hongsheng Guo[2], Li Lu[3], Xuhui Zhang[4,5,6,7], Yan Zhang[5,6], Zhiwei Hao[1], Xiaobing Lu [4,5,6]*

**1** CPOE Research Institute of Engineering Technology, Tianjin, China, **2** CNPC Offshore Engineering Company Limited, Beijing, China, **3** Hubei Communications Planning and Design Institute Co., Ltd., Wuhan, China, **4** Guangdong Aerospace Research Academy, Guangzhou, China, **5** Institutes of Mechanics, Chinese Academy of Sciences, Beijing, China, **6** School of Engineering Science, University of Chinese Academy of Sciences, Beijing, China, **7** Key Laboratory for Mechanics of Fluid Solid Coupling System, Institutes of Mechanics, Chinese Academy of Sciences, Beijing, China

* xblu@imech.ac.cn

## Abstract

The imbibition of water into the pores of tight oil/gas reservoir can displace the oil/gas out. Thus it is an important method to improve the recovery efficiency of tight shale gas and oil. This paper investigated the influence of four main dimensionless parameters on the spontaneous imbibition based on a pores distribution of a real shale sample. The results show that the connectivity has the greatest impact on the average imbibition velocity while the impact of the contact angle is the smallest. The capillary number has the greatest impact on the oil displacement efficiency. The impact of main factors on imbibition and displacement is not monotonic, but rather a combination of these factors.

## 1. Introduction

Imbibition is a common phenomenon in dense rock and soil. The imbibition of water into the pores of rock and soil can displace the oil out and sometimes cause deterioration of the properties of the rocks, leading to the destruction. Therefore, imbibition is an important phenomenon driven by capillary pressure in low and ultra-low permeability rock and soil with small pores and throats [1].

In order to study the relationship between the imbibition length and time, many imbibition experiments on tight core have been conducted [2]. However, during experiments, it is difficult to monitor the movement of the fluid inside the sample. Advanced simulation tools such as Lattice Boltzmann Method (LBM) can provide an economical and effective way to study the effects of physical and mechanical properties and complex geometry [3].

**Data availability statement:** The data that support the findings of this study are available within the article and its Supporting Information files.

**Funding:** This work was supported by the National Natural Science Foundation of China (U2344223, 12302516,11872365), the CNPC New Energy Key Project (2021DJ4902); and the High-level Innovation Research Institute Program of Guangdong Province(No.2020B0909010003). The funders had no role in study design, data collection and analysis, decision to publish, or preparation of the manuscript.

**Competing interests:** The authors have declared that no competing interests exist.

LBM originated from molecular dynamics and was enhanced through statistical mechanics. It has been widely applied in computational fluid dynamics [4–7]. As a kind of mesoscale method, LBM can capture the micro physical phenomena within the manageable and effective use of computer resources, and transform them into macroscopic parameters.

In recent years, LBM has been used for the analysis of ultra-low permeability reservoir, such as the analysis on the influence of characteristic size, boundary layer, initial water saturation, etc. on imbibition [8–10]. Bakhshian et al. [11] used LBM to investigate the effects of interface motion and roughness on the imbibition of porous media under complex pore fracture structures. Hatiboglu et al. [12] used LBM to simulate the co-current and countercurrent imbibition in porous fractural rocks. Wang et al. [13] presented a pseudopotential-based multiple-relaxation-time LBM. This model can study the multicomponent flows with different molecular weights, different viscosities and different Schmidt numbers. Zhang et al. [14] studied the pore scale dynamics of imbibition in heterogeneous sandstone samples using LBM. They have not studied the dynamics of imbibition in other types of rocks. Zhou et al. [15] summarized the numerical simulation methods and research progress of spontaneous imbibition at the micro pore scale, and compared the advantages and disadvantages of various methods. Wang et al. [16] studied the imbibition in nanoporous media using LBM. Cai et al. [17] discussed the advantages and disadvantages of the classic Lucas Washburn equation, as well as its development and applications. Cai et al. [18] studied the interface dynamics and fluid-fluid interactions during imbibition of porous rocks by introducing pore topology measurement. They found the reason of unstable inlet pressure, mass flow rate, and interface curvature.

Most of the previous work was based on the assumption in the Lucas Washburn equation, where porous media was conceptualized by a bundle of linear capillaries [19]. However, it cannot describe the interaction among pores and fractures and the non-uniformity in tight oil/gas reservoirs, which has a significant impact on the imbibition process and the distribution of remaining oil and gas.

This paper focuses on the impact and quantitative analysis of the main factors. The widely used color LBM in the analysis of multiple phases flow was adopted in this study. The code was written according to the mathematical model of color LBM described in Section 2 first. Then the numerical simulation was built based on the pore-throat size distribution of a real shale core measured by the author Lu Li [20]. Firstly, the main factors were determined through dimensional analysis, and then the impact of the main dimensionless factors was calculated and analyzed through LBM. The fitting relationship between the displacement efficiency and the main factors such as contact angle and connectivity was established.

## 2. Simple introduction of the color LBM

Leclaire et al. [21] presented a kind of color LBM based on the model of Ortona et al. [22]to analyze the non-miscible two-phase flow. This paper used the D2Q9 discrete format [5] to simulate a two-phase fluid. The evolution equation is implemented through the following steps.

## 2.1. Collision operator $\Omega_{ki}^1$

$\Omega_{ki}^1$ for single phase in this model is the same as that in BGK model (standard D2Q9 model)(Fig 1), which can be expressed as follows [23]:

$$\Omega_{ki}^1 = f_{ki}(x,t) - \omega_k \left( f_{ki}(x,t) - f_{ki}^e(x,t) \right) \tag{1}$$

in which $\omega_k$ is relaxation factor, $f_{ki}^e$ is a distribution function in equilibrium state, which is as follows:

$$f_{ki}^e = \rho_k \left\{ \phi_i^k + \omega_i \left[ 3c_i \cdot u + \frac{9}{2} (c_i \cdot u)^2 - \frac{3}{2} u^2 \right] \right\} \tag{2}$$

in which $u$ is the total local speed of the two phases, $\omega_i$ is the weights.

## 2.2. Perturbation operator $\Omega_{ki}^2$

Applying perturbation operators to interfacial tension in color models. The degree of color change is described by the color gradient $F(x)$ perpendicular to the interface. $F(x)$ is adopted as in the following computation:

$$F(x) = \sum_i c_i \left[ \rho_r (x + c_i) - \rho_b (x + c_i) \right] \tag{3}$$

in which "i" represents either "r" or "b", the two fluids, "r" indicates red phase and "b" indicates blue phase.

The perturbation operator $F(x)$ is expressed as follows:

$$\Omega_{ki}^2(x,t) = \frac{A_k}{2} |F| \left[ W_i \frac{(F \cdot c_i)^2}{|F|^2} - B_i \right] \tag{4}$$

in which $A_k$ is the interfacial tension, $B_i$ is a parameter.

Although the perturbation operator $\Omega_{ki}^2$ can reflect the interfacial tension, it cannot guarantee that these two phases do not invade each other. Therefore, the recolor operator $\Omega_{ki}^3$ is introduced to separate the two phases.

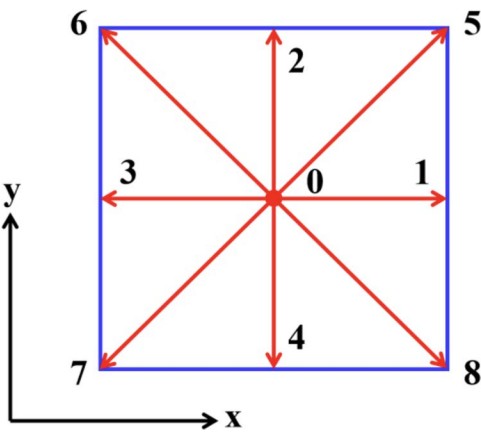

**Fig 1. Model of D$_2$Q$_9$ (0~8 indicate the nine node numbers).**

## 2.3. Recolor operator $\Omega_{ki}^3$

Gunstensen presented the following distribution function after the evolution of recolor operator [24]:

$$W\left(f_{ri}'' + f_{bi}''\right) = \max\left[\sum_i \left(f_{ri}'' - f_{bi}''\right) c_i\right]$$

(5)

However, Gunstensen model [24] is complex in computation, thus some simplified methods are presented. For example, Sebastien-Leclaire [25,26] provided the following operator:

$$\Omega_{ri}^3\left(f_{ri}\right) = \frac{\rho_r}{\rho} f_i + \beta \frac{\rho_r \rho_b}{\rho^2} \cos\left(\varphi_i\right) \sum_k f_{ki}^e\left(\rho_k, 0, \alpha_k\right)$$

(6)

$$\Omega_{bi}^3\left(f_{bi}\right) = \frac{\rho_r}{\rho} f_i - \beta \frac{\rho_r \rho_b}{\rho^2} \cos\left(\varphi_i\right) \sum_k f_{ki}^e\left(\rho_k, 0, \alpha_k\right)$$

(7)

in which β is a free parameter, $\varphi_i$ is the angle between the color gradient $F(x)$ and the sound velocity $c_i$ at lattice points. The distribution function $f_{ki}^e$ computed by using the parameters $\alpha_k$, k indicates r and b phase and $u = 0$.

## 3. Dimensional analysis

Consider the problem as a plane problem, the parameters of water and oil are as follows: material parameters: density of oil $\rho_o$, viscosity of oil $\mu_o$, density of water $\rho_w$, viscosity of water $\mu_w$, interfacial tension coefficient σ, wetting angle θ, permeability k, porosity ϕ; geometrical parameters of pores: due to the pore size distribution of the selected core sample is close to a normal distribution, the relevant parameters here are the average pore throat radius R and the standard deviation α; others: percolation probability P, the gravity acceleration g.

Considering that the influence of gravity is small in the early and middle stages of imbibition, especially in the horizontal imbibition, it is not considered in the calculation; After the distribution of pore scale is determined, the total porosity is also determined, and there is a relationship between permeability, porosity, and pore radius ($R^2 = \frac{8k}{\phi}$). Thus to normalize the other parameters by using $\rho_w$, $\mu_w$, R and considering $\frac{\rho_o}{\rho_w}$ is basically thought as a constant, Then the displacement efficiency η and the displacement velocity $\dot{h}$ can be expressed as:

$$\left\{\begin{array}{c} \eta \\ \frac{\dot{h}}{u^*} \end{array}\right\} = f\left(\frac{\mu_o}{\mu_w}, \frac{\mu_w u^*}{\sigma}, \theta, \frac{k}{R^2}, \alpha, P\right)$$

(8)

in which $u^* \sim \frac{\mu_w}{\rho_{wR}} \sim \dot{h}$ is the characteristic velocity $\frac{\mu_o}{\mu_w}$ is the oil-to-water viscosity ratio, $\frac{\mu_w u^*}{\sigma}$ is Ca, θ is the wetting angle, $\frac{k}{R^2}$ is the dimensionless permeability, $\alpha$ is the standard deviation of pore radius distribution, P is the percolation probability of pores.

## 4. Numerical simulation

### 4.1. Simulation scheme

The following calculations will focus on the four main dimensionless parameters Ca、 θ、 $\frac{\mu_o}{\mu_w}$、 P) for this problem based on the dimensional analysis in the previous section. Firstly, a set of baseline value (contact angle 36°, water-to-oil viscosity ratio 0.2, capillary number 0.0224, percolation probability of pores 100%) was calculate, and then the impact of each parameter change by taking 0.8, 0.9, 1.1, and 1.2 times the baseline value were calculated, while keeping the other values constant.

### 4.2. Numerical model and parameter selection

Due to the limitations of computation time and computer capacity, the numerical model is considered as planar imbibition, and the measured pore size distribution of a real shale core was selected to establish the model. The left side is the inlet end for water entering, the right side is the outlet end for oil expulsion, and the upper and lower boundaries are impermeable conditions.

#### 4.2.1. Pore size distribution of a real core.
The size distribution of pores adopted in the numerical model is based on the mercury intrusion test of a real shale core and is approximately normal: the average value is $9.384 \times 10^{-2} \mu m$ and the standard deviation is $2.276 \times 10^{-2} \mu m$. Taking this model as the baseline example, to investigate the influence of dimensionless parameters.

#### 4.2.2. Baseline model example and numerical results.
*4.2.2.1. Model establishment and parameter setting:* The baseline model is shown in Fig 2, where green represents the water phase, red represents the oil phase, and blue represents the solid phase. The upper and lower boundaries of the model are solid boundaries without seepage, and the left and right boundaries are free boundaries with equal pressure. The pore network zone is 7.44 μm long, 3.435 μm high, with a short pore length of 315 nm and an average pore diameter of 93.84 nm. There is a free flow field with no internal obstacles on both the left and right sides of the pore network area. Water is sucked into the pore network from the left side due to capillary force, and oil is displaced out of the right side. The values of the other parameters are as follows: density of water 1000 kg/m³, dynamic viscosity of water 1mPa*s, density of oil 850 kg/m3, dynamic viscosity of oil 5mPa*s, contact angle of 36°, interface tension coefficient of 0.54mN/m [27]. Fig 3 shows the variation of two-phase mass over time. It can be seen that the oil displacement rate is almost linear, the displacement efficiency is about 85%.

*4.2.2.2. Certification of the model:* To certify the numerical model, we first compared the results of baseline examples and experimental results and then compared the results with Young-Laplace equation.

Fig 4a shows the comparison between the numerical results of the baseline examples and the experimental results of a real shale core [28]. The data is converted into dimensionless characteristic time ( $t_D = t \sqrt{\dfrac{k}{\phi} \dfrac{\sigma}{\sqrt{\mu_o \mu_w}} \dfrac{1}{L_c^2}}$ ). The points in the figure represent the experimental data, and the solid line represents the numerical data. It can be seen that the simulation results are quite close to the experimental values. The correlation coefficient between the numerical value and the experimental value reaches 96%.

To compare with the **Y**oung-Laplace equation, a 100 × 100 square lattice is set with a bubble of red fluid centered in the middle of the zone. The pressure difference inside and outside the bubble is computed. The parameters used in computation are $\rho_r = 1.0$, $\rho_b = 1.0$, $A_r = A_b = 0.001$, $\alpha_r = 0.1$. Fig 4b shows the computed pressure difference against $\frac{1}{R}$, in which R is the radius. The dots are the computed data, the solid line is a linear fit (theoretical data). The slop of the line gives the surface tension. The computed data are shown to be agree well with the theoretical data.

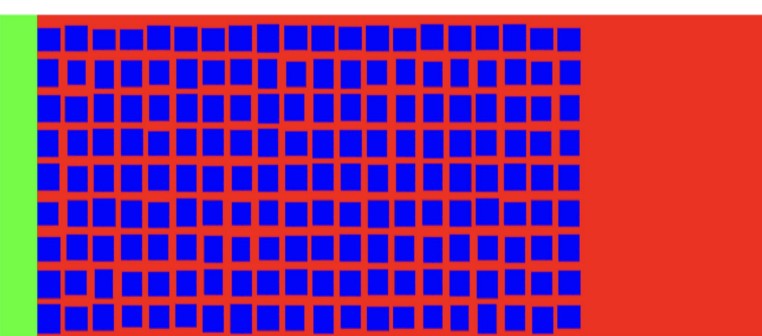

**Fig 2. Numerical model of baseline model.**

## 4.3. Numerical results

In this section, the influences of main dimensionless parameters on the oil displacement efficiency and displacement rate are investigated. The non-uniform development of the infiltration front has also been analyzed.

### 4.3.1. Effects of capillary number.

*4.3.1.1. Distribution of oil and water during imbibition:* From the changes in the distribution of oil-water over time (Fig 5), it can be seen that the displacement front is relatively uniform at the beginning, while the front gradually becomes uneven over time, and the displacement rate increases at the upper and lower boundaries. There is residual oil behind the front. At the same time, it should be noted that the co-current imbibition and countercurrent imbibition are processed simultaneously, but mainly is the co-current imbibition. In the case that the capillary number is 80% of the baseline model

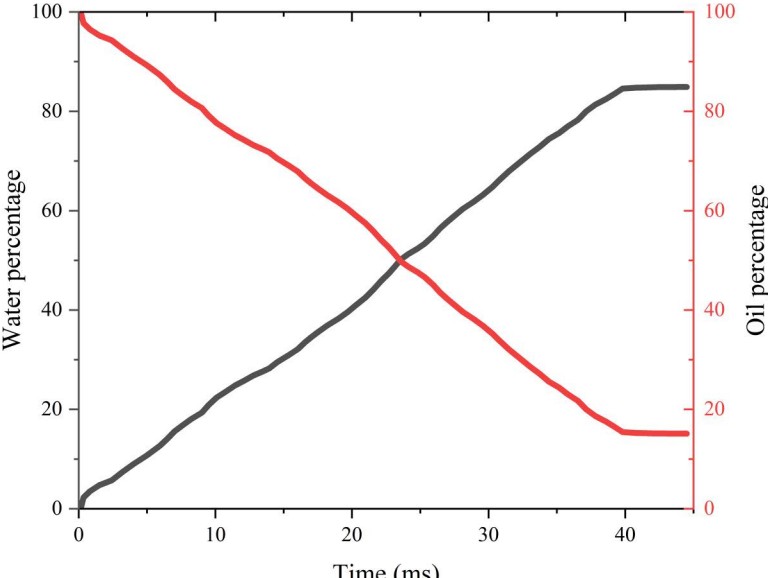

**Fig 3. Variation of oil-water mass with time during imbibition process.**

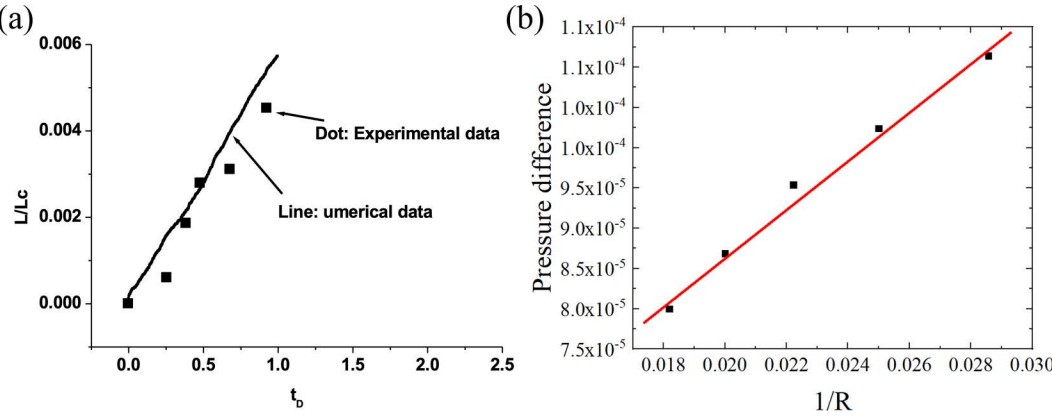

**Fig 4. Comparison between simulation results and experimental results of baseline examples and Young-Laplace equation.** (a) Compare with experimental results of baseline examples (b) Compare with Young-Laplace equation.

(Fig 5a), a large area of residual oil is concentrated in the middle of the region at the end of displacement. The reason is of the upper and lower boundary effects, where the imbibition near the boundary develop rapidly. When the imbibition front of these two parts exceeds the middle oil layers, they are enveloped and cannot be displaced.

When the capillary number is 90% of the baseline model, the displacement near the upper boundary is faster, while it becomes slower and slower near the lower boundary and the middle region (Fig 5b). Finally, more residual oil is concentrated in the middle and lower parts. This may be due to the fact that as the capillary number increases, the pore scale distributed near the upper boundary just becomes larger, coupled with boundary effects, resulting in a faster imbibition rate in this area than in other areas. When the capillary number is 110% of the baseline model, most of the oil is displaced out and less is retained. In other words, the displacement efficiency is high (Fig 5c). When the capillary number is 120% of the baseline value, Most of oil is displaced out except for some point-like residual oil, and a small piece of residual oil in the lower right corner (Fig 5d). That means, the oil displacement efficiency. is high.

***4.3.1.2. The variation curve of two-phase mass with time:*** Here, the average imbibition length is determined by dividing the water phase mass by the product of water density and cross-sectional area. It can be seen from the variation of water with time that the displacement front fluctuates with time, but the imbibition rate varies with the number of capillaries. When the capillary numbers are 0.8, 0.9, 1.1, and 1.2 of the baseline value, the average displacement speeds are 8.72 mm/s, 12.14 mm/s, 13.18 mm/s, and 7.67 mm/s, respectively (Fig 6). When the capillary number increases or the surface tension coefficient increases, on the one hand, the driving force of imbibition increases, and on the other hand, the resistance increases. Therefore, the influence of capillary number on imbibition and displacement is complicated.

When the capillary numbers are 0.8, 0.9, 1.1, and 1.2 of the baseline value, the final oil displacement efficiencies are 87.22, 61.03, 87.77, and 80.56, respectively. As the capillary number increases, the residual oil increases also, but not a completely monotonic relationship (Fig 6). This indicates that the relationship between the oil displacement efficiency and capillary number is not single-valued relationship, but the result of the combined action of the main parameters.

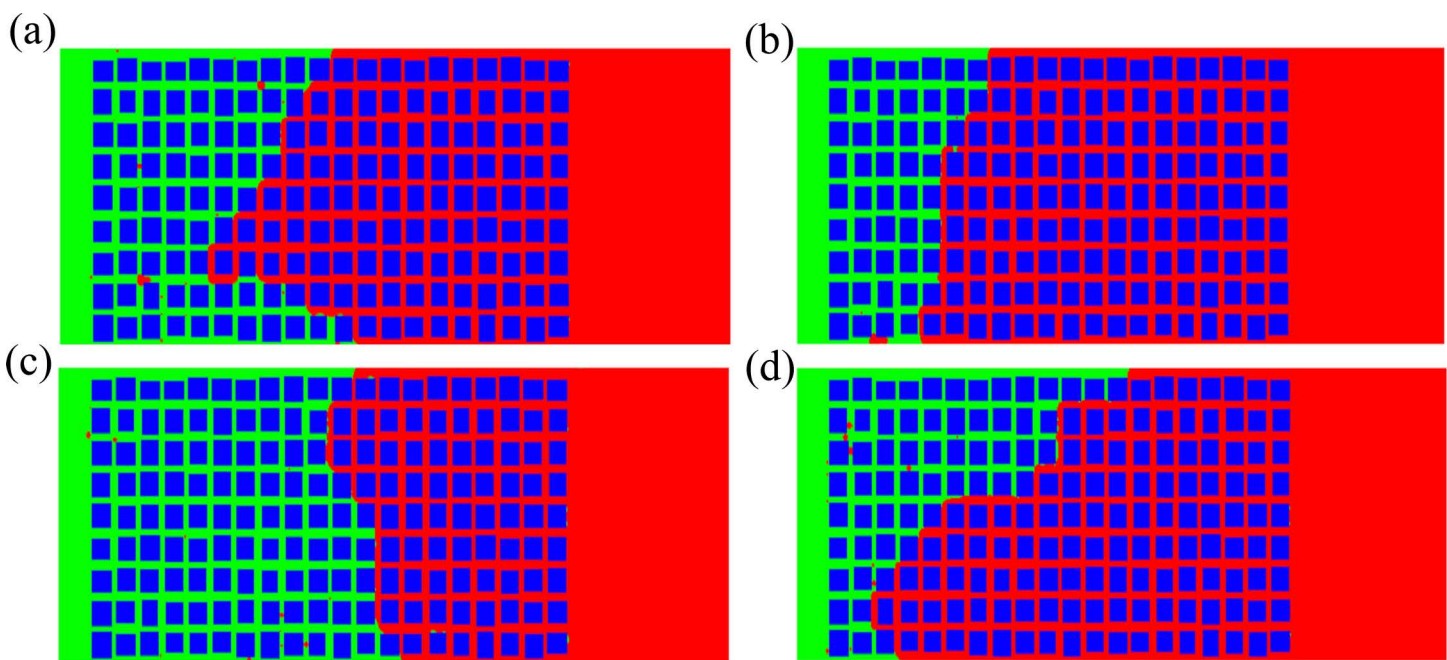

**Fig 5. When the capillary number changes, the oil-water distribution changes with time.** (a) 200μs (80%) (b) 100μs(90%) (c) 300μs (110%) (d) 200μs (120%).

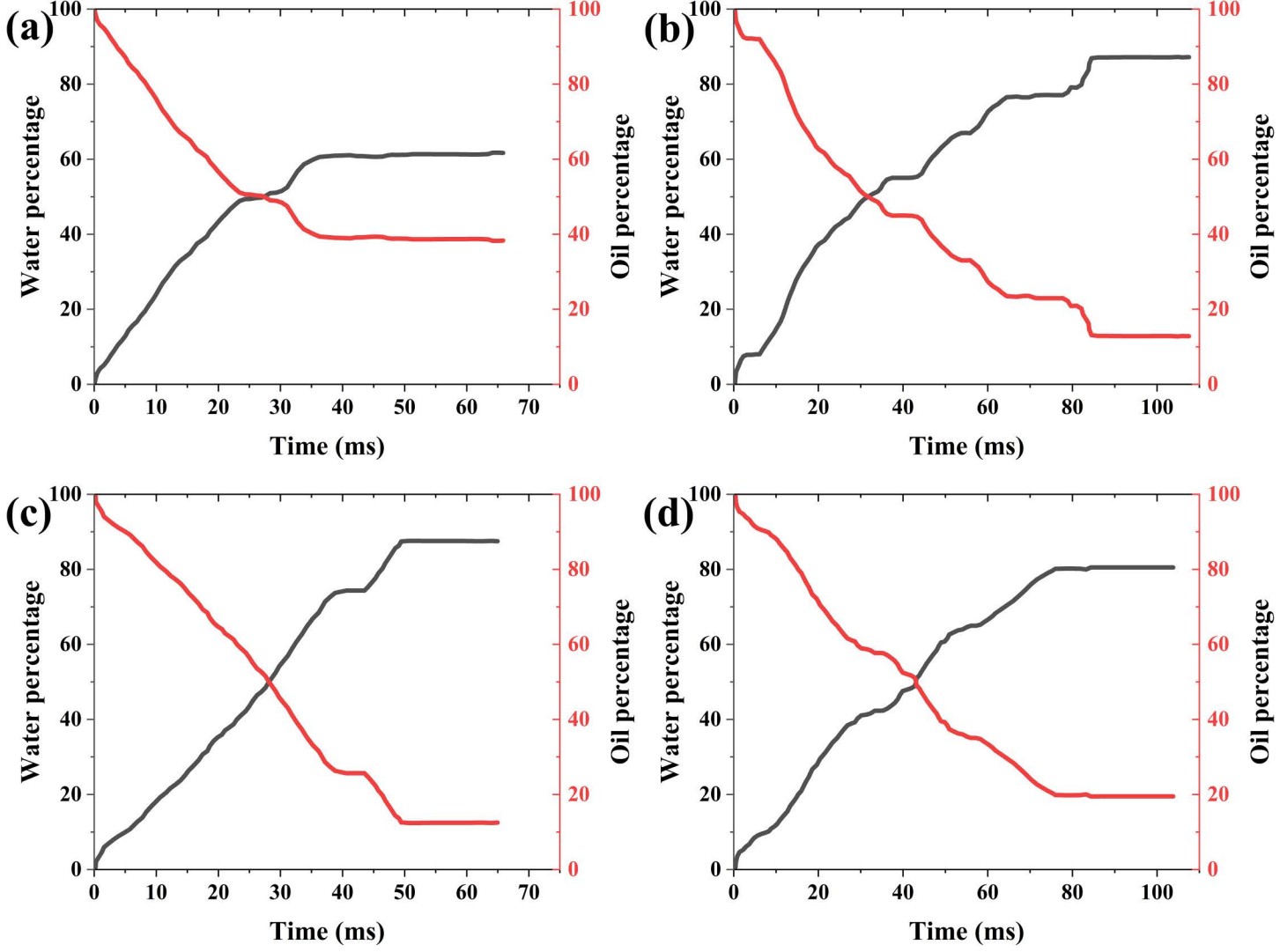

**Fig 6. Variation of oil-water two-phase mass with time and capillary number.** (a) 80% of the baseline value (b) 90% of the baseline value (c) 110% of the baseline value (d) 120% of the baseline value (The red line indicates the oil mass percentage, the black line indicates the water mass percentage. They have the same meaning in the following figures.).

To plot a curve between the displacement efficiency and capillary number, it can be seen that the placement efficiency increases with capillary number in a near parabolic (Fig 7).

### 4.3.2. Effects of contact angle.

*4.3.2.1. Development of imbibition front and distribution of oil and water:* It can be seen from the distribution of two-phase fluids during imbibition that when the contact angle is 80% of the baseline value, the development of imbibition at the upper and lower parts is fast, and at the middle part is slow, when the displacement interface at the upper and lower parts exceeds the middle of the computing zone, they are connected and seal off a portion of the middle area and form the residual oil, resulting in the presence of small pieces of residual oil in the middle area(Fig 8a). The overall displacement efficiency is high. When the contact angle is 90% of the baseline value, the upper part develops faster than the lower part, and finally a large amount of residual oil is located in the lower part (Fig 8b). When the contact angle is

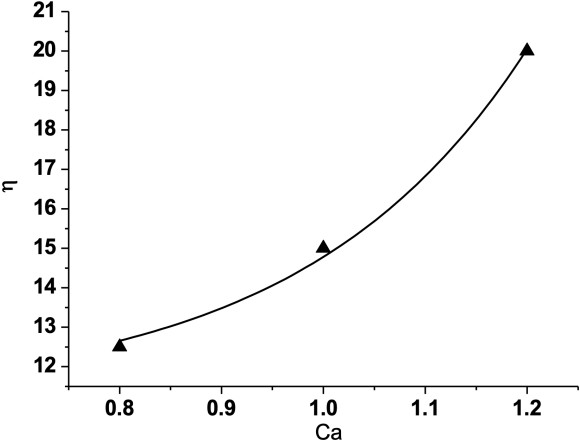

**Fig 7. Relation between displacement efficiency and Ca.** (The fitted curve: $\eta = 11.22 + 0.038\mathrm{e}^{5\mathrm{Ca}}$).

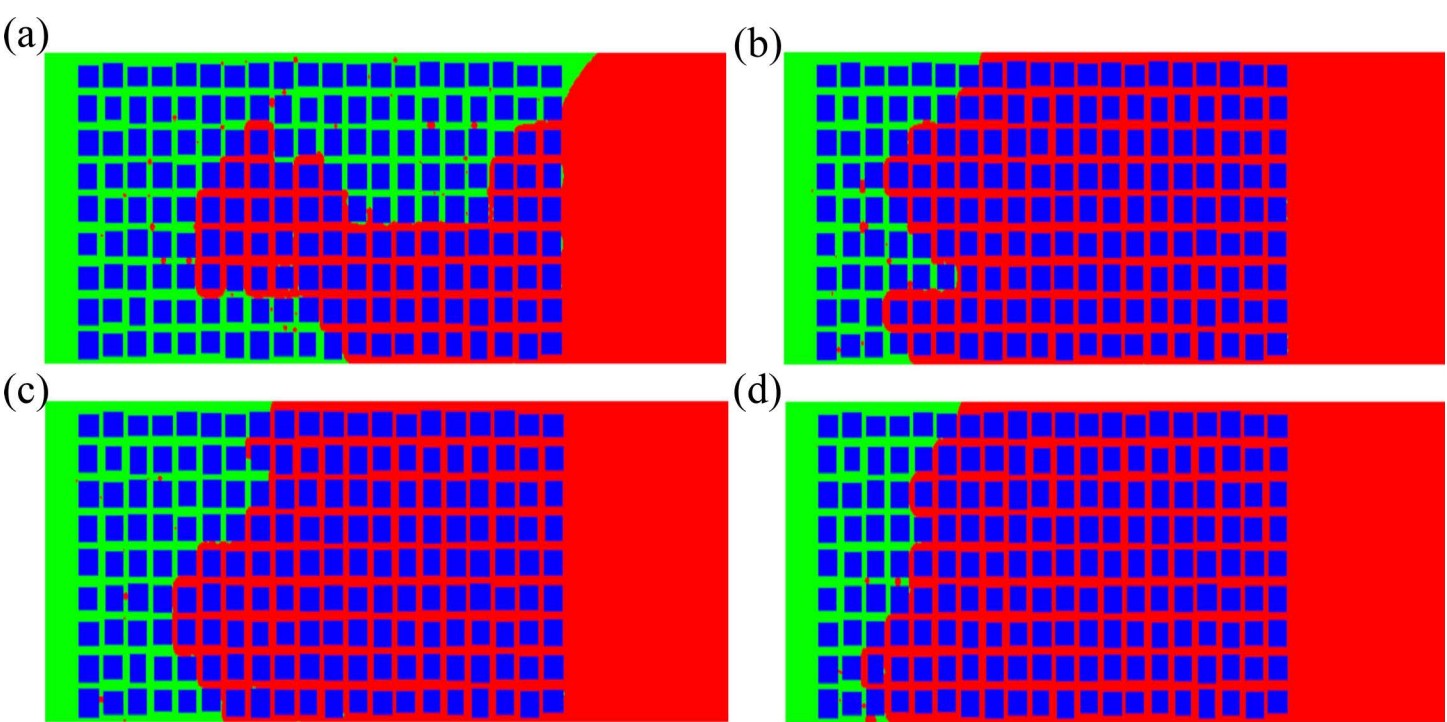

**Fig 8. Whencontact angle changes, the oil-water distribution changes with time.** (a) 300μs(80%) (b) 100μs (90%) (c) 100μs (110%) (d) 100μs (120%).

110% of the baseline value, the imbibition in the upper and lower parts is fast, while the development in the middle part is slow. However, with the development of imbibition, the majority of the oil in the middle is also displaced, resulting in less residual oil (Fig 8c). When the contact angle is 120% of the baseline value, there is still much residual oil in the middle and lower parts (Fig 8d)

***4.3.2.2. The variation of two-phase mass with time:*** From the curve of two-phase mass over time during imbibition, it can be seen that the water gradually fluctually enters, while the oil is gradually displaced, but the fluctuation amplitude is relatively small. When the contact angles are 0.8, 0.9, 1.1, and 1.2 of the baseline value, the displacement efficiencies are 89.96%, 53.54%, 98.44%, and 41.49%, respectively (Fig 9). The displacement efficiency and average displacement velocity do not vary monotonically with the change of contact angle (Fig 10).

The contact angle reflects the wettability of the medium. The larger the contact angle, the lower the wettability, the smaller the capillary force, and the larger the oil droplet size, all of which are more unfavorable for displacement. The smaller the contact angle and the better the wettability, the smaller the droplet size and the more favorable it is for droplet displacement.

**4.3.3. Effect of viscosity ratio.** ***4.3.3.1. The variation of oil-water distribution with time:*** When the viscosity ratio is 80% of the baseline value, the displacement development is more uneven, and the imbibition development in the lower zone is faster. Unlike the changes in the previous factors, when the lower displacement is about to end, the displacement

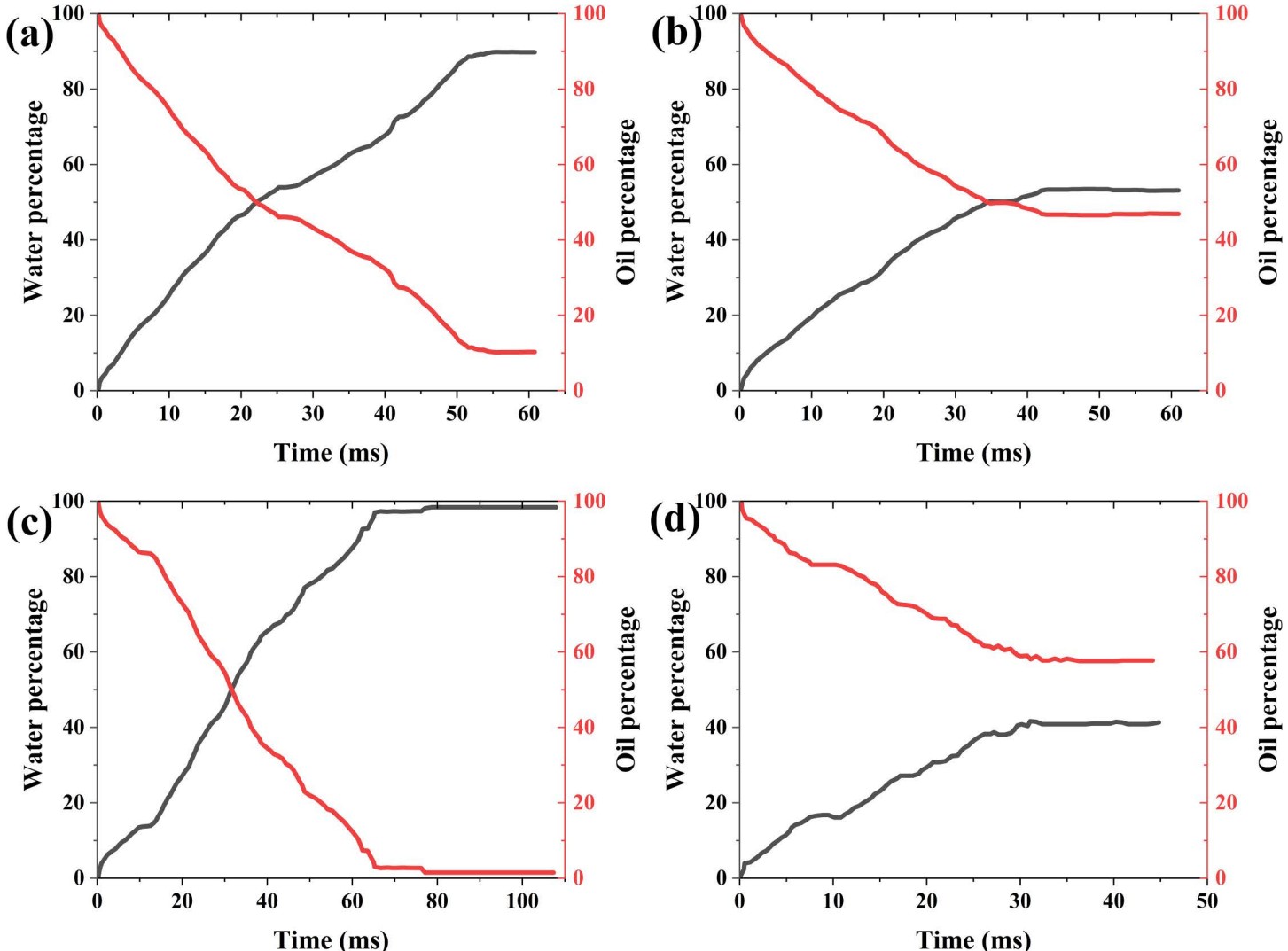

**Fig 9. Variation of oil-water two-phase mass with time and contact angle.** (a) 80% of the baseline value (b) 90% of the baseline value (c) 110% of the baseline value (d) 120% of the baseline value.

gradually develops towards the upper region. In this case, the main mode is co-current imbibition, but there is also weak countercurrent imbibition. The overall displacement efficiency is relatively high, and in the end, most of the oil is displaced out (Fig 11a). When the viscosity ratio is 90% of the baseline value, the non-uniformity of imbibition is more severe than when the viscosity ratio is 80% of the baseline value. The imbibition development near the upper boundary is significantly greater than that in the middle and lower parts, forming a dominant channel in the upper part, similar to channeling flow. As a result, only a small part of the oil is displaced out in the end (Fig11b). When the oil dynamic viscosity is 110% of the baseline value, the development of imbibition is also very uneven, but the oil displacement efficiency is still relatively high

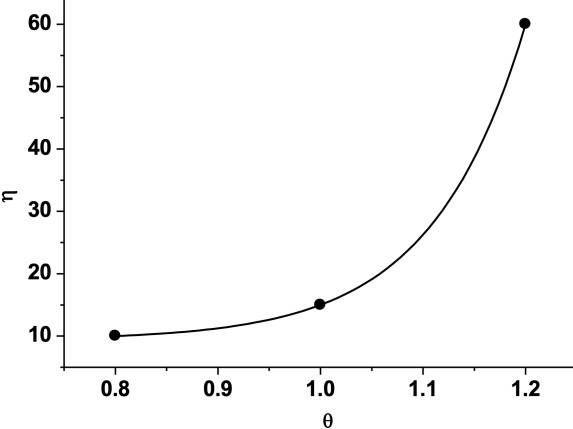

**Fig 10. Changes of displacement efficiency of oil with contact angle.** The fitted curve is $\eta = 9.38 + 10^{-3}\mathrm{e}^{11\theta}$.

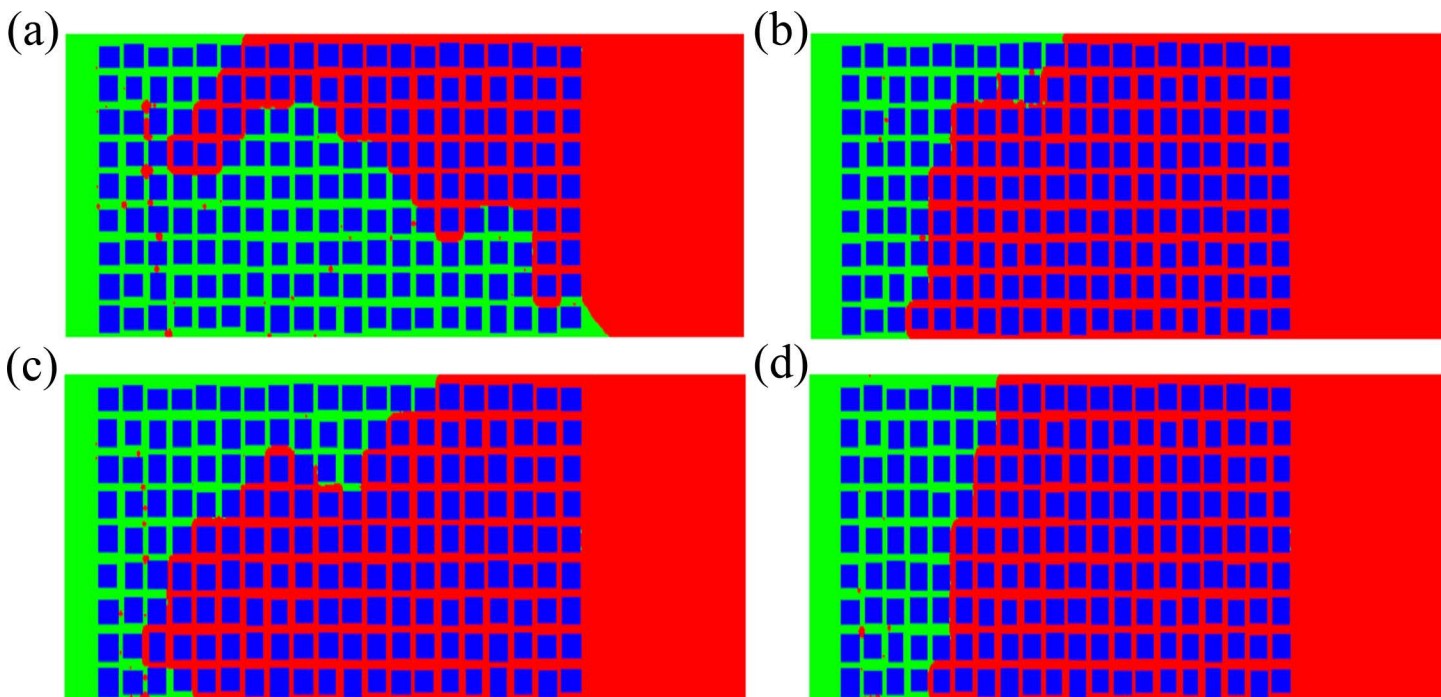

**Fig 11. When the viscosity ratio changes, the oil-water distribution changes with time.** (a) 200μs (80%) (b) 100μs (90%) (c) 150μs (110%) (d) 100μs (120%).

in the end. There are sheet residual oil in the middle region, and point like residual oil in other locations ([Fig11c]). When the viscosity ratio is 120% of the baseline value, a large amount of residual oil is formed in the middle and lower parts ([Fig 11d]). The reason is that the increase of viscosity ratio is equivalent to an increase in the viscosity of the oil, resulting in a significant increase in imbibition resistance and a slow development of imbibition. Meantime, the instability of the interface increases, making it easier to form advantageous channels and leading to the occurrence of channeling flow.

Generally, the higher the oil-water viscosity ratio, the greater the resistance to oil displacement by imbibition, and the more likely it is to cause fingering, resulting in lower oil recovery efficiency. On the other hand, because capillary force is the main driving force for spontaneous imbibition, and the size of the oil-water viscosity ratio will affect capillary force, thereby changing the speed and time of imbibition.

**4.3.3.2. The change of two-phase mass with time:** The variation characteristics of displacement speed and residual oil with time and viscosity ratio are shown in [Fig 12]. As can be seen, with the development of imbibition, the water gradually

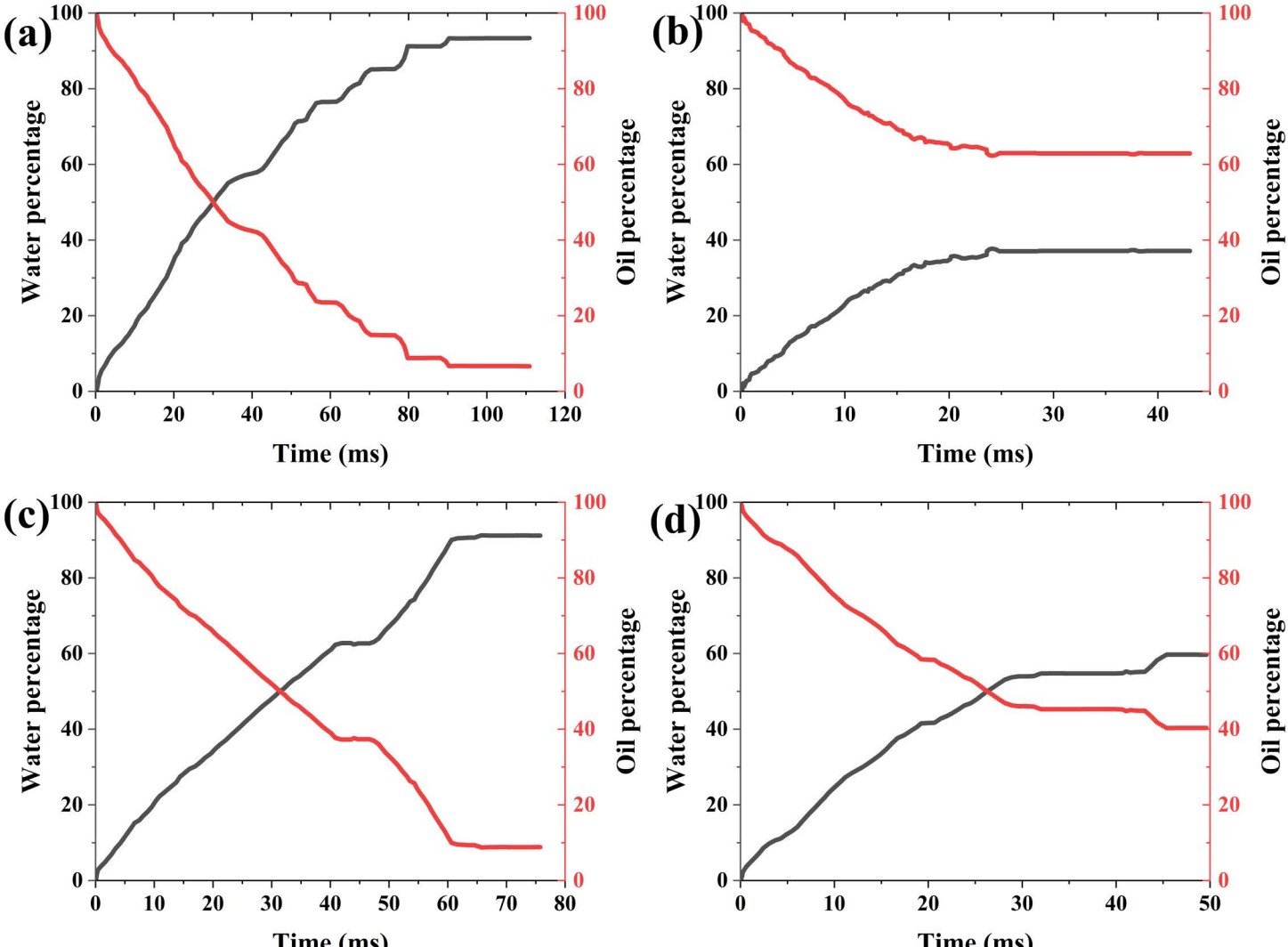

**Fig 12. Variation of oil-water two-phase mass with time and viscosity ratio.** (a) 80% of the baseline value (b) 90% of the baseline value (c) 110% of the baseline value (d) 120% of the baseline value.

enters and the oil is gradually displaced out, both processes are characterized by significant fluctuated development. When the viscosity ratios are 80%, 90%, 110%, and 120% of the baseline value, the final displacement efficiencies are 93.39%, 37.66%, 91.25%, and 59.78%, respectively. It can be seen that there is no monotonic relationship between displacement efficiency and viscosity ratio (Fig 13).

### 4.3.4. Effect of connectivity.

To investigate the influence of connectivity on the imbibition and displacement process, considering the square grid used in the simulation, the percolation threshold is about 50%. Therefore, only a few operating conditions with connectivity greater than 50%, such as 60%, 70%, 80%, and 90%, will be analyzed, and the remaining parameters are the same as the baseline value.

*4.3.4.1. The variation of oil-water distribution with time:* When the connectivity is 60%, the development of imbibition is very slow. Although it is not completely displaced in the end, the imbibition front no longer develops (Fig 14a), indicating that the adopted connectivity has not yet reached the percolation threshold of the set grid. There are more residual oil in the imbibition zone, indicating lower displacement efficiency. When the connectivity is 70%, the displacement speed is faster than that at 60%, and the displacement length is longer than the former, indicating that the increase in connectivity promotes the development of imbibition. At 800 μs, the displacement length of the former is less than 50% of the total length of the region, while at 350 μs, the displacement length of the latter has reached 60% of the total length of the region. However, the two-phase interface of the latter is more uneven (Fig 14b). When the connectivity is 80%, the imbibition rate is faster than that when the connectivity is 60% and 70%, and the displacement front has reached the other end under this condition (Fig 14c). There is a significant countercurrent imbibition. However, there is a large amount of residual oil in the displacement zone, indicating that the displacement is too fast and the process is prone to instability. When the connectivity is 90%, the displacement rate is faster than the previous cases. At 300 μs, displacement has been completed, while when the connectivity is 80%, displacement completion requires 1000 μs. However, the development of displacement is more uneven, with a large amount of residual oil present in the middle and lower parts (Fig 14d). This indicates that although the increase of connectivity promotes imbibition rate, it is also more likely to cause interface instability and increase residual oil.

*4.3.4.2. Changes in two-phase mass over time:* From the changes in the quality of oil-water with time and connectivity, it can be seen that there are fluctuations in the imbibition rate. When the connectivity is 60% and 70% respectively, the amount of water entering the region is small and the development rate is slow. In the latter two cases, the imbibition rate is significantly accelerated, but the residual oil at 80% connectivity is higher than that at 90% connectivity, and lower than the values at 60% and 70% connectivity (Fig 15), showing the different impact trends of connectivity on imbibition rate and residual oil.

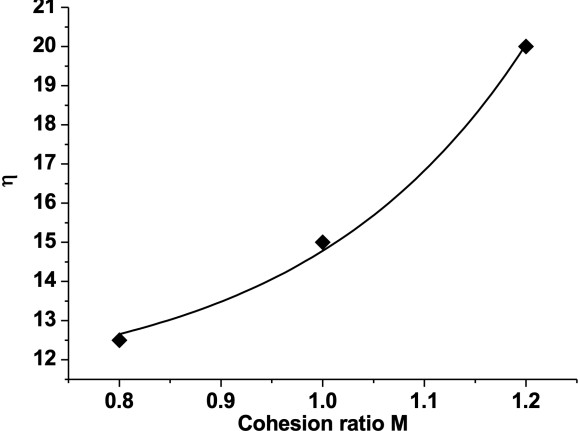

**Fig 13. Effects of cohesion ratio on oil-displacement efficiency. The fitted curve is $\eta = 11.22 + 0.038e^{5M}$.**

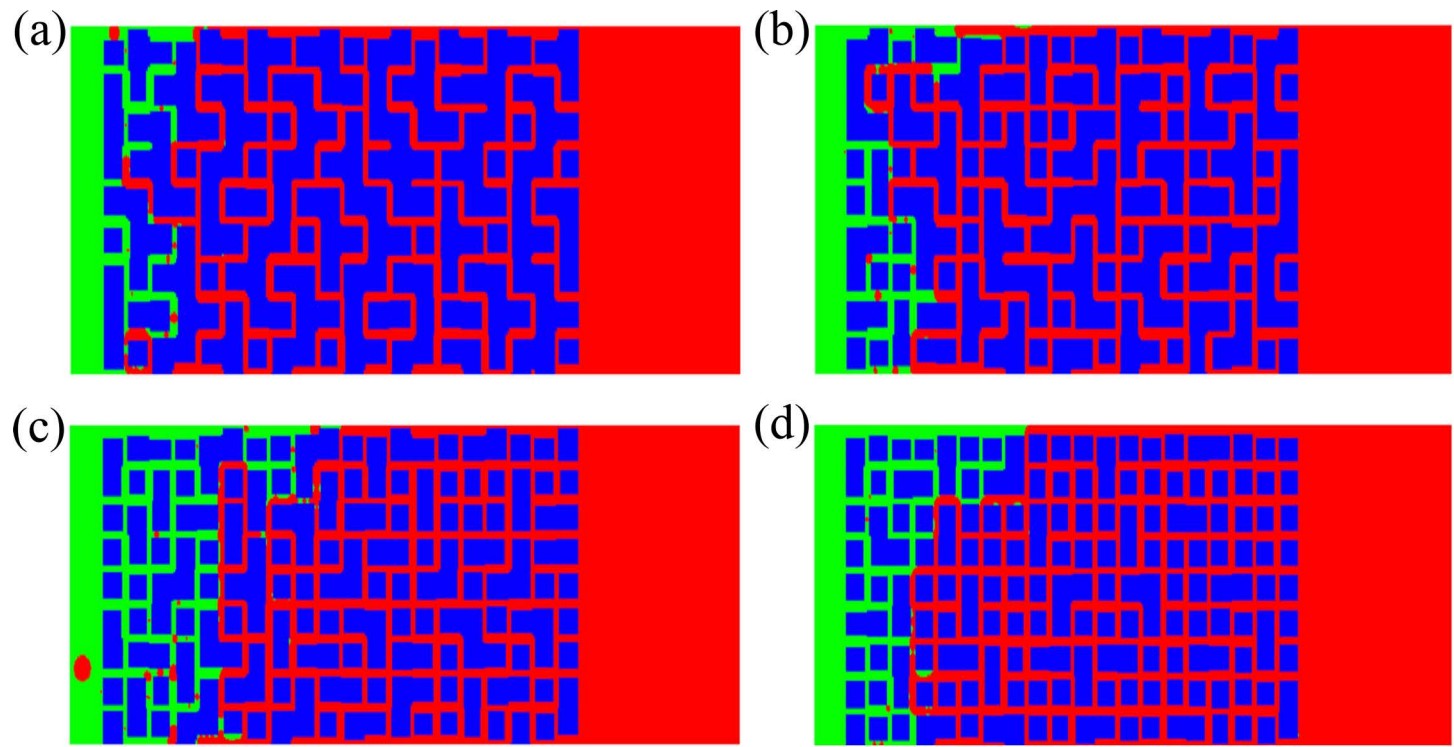

**Fig 14. At the connectivity is 90%, the oil-water distribution changes with time.** (a) 200μs (60%) (b) 100μs (70%) (c) 200μs (80%) (d) 100μs (90%).

The impact of various factors on imbibition and displacement is not monotonic, but rather a combination of these factors. This can also be seen from Fig 16. This graph shows the relative variation of oil displacement efficiency with respect to the relative variation of the four dimensionless parameters. It can be seen that under the calculation conditions in this section, there is a peak in displacement efficiency, located at the baseline value. On the one hand, it reflects the non-monotonicity between the average imbibition rate and displacement efficiency and various parameters, and on the other hand, it shows the difference in the relative changes in the average imbibition rate and displacement efficiency caused by the relative changes in each parameter, that is, the relative magnitude of the influence of various factors.

### 4.4. Discussions

From the above results we can see that the development in the small throats is faster than that in the large throats. The path of displacement is strongly affected by the interaction among the throats (i.e., the capillary forces in these throats). According to L-W equation, the imbibition velocity in large throats is larger than that in small throats. However, the numerical results show that the imbibition velocity in small throats are larger because of the larger capillary forces which induce the displacement develops from the small throats to large throats. Therefore, in a pore-fracture network, the interaction among throats/fractures must be considered and the development of displacement in the throats with different radii must be variable. That means, the imbibition in the throats with small radius and small volume will first finish. The apparent imbibition velocity and imbibition mass will decrease gradually, which is coincidence with the experimental results [24].

Because of the non-uniform distribution of pore size, the imbibition interface does not develop smooth. In some cases, channeling path forms and so causes the decrease of displacement efficiency. The reason is that the instability of

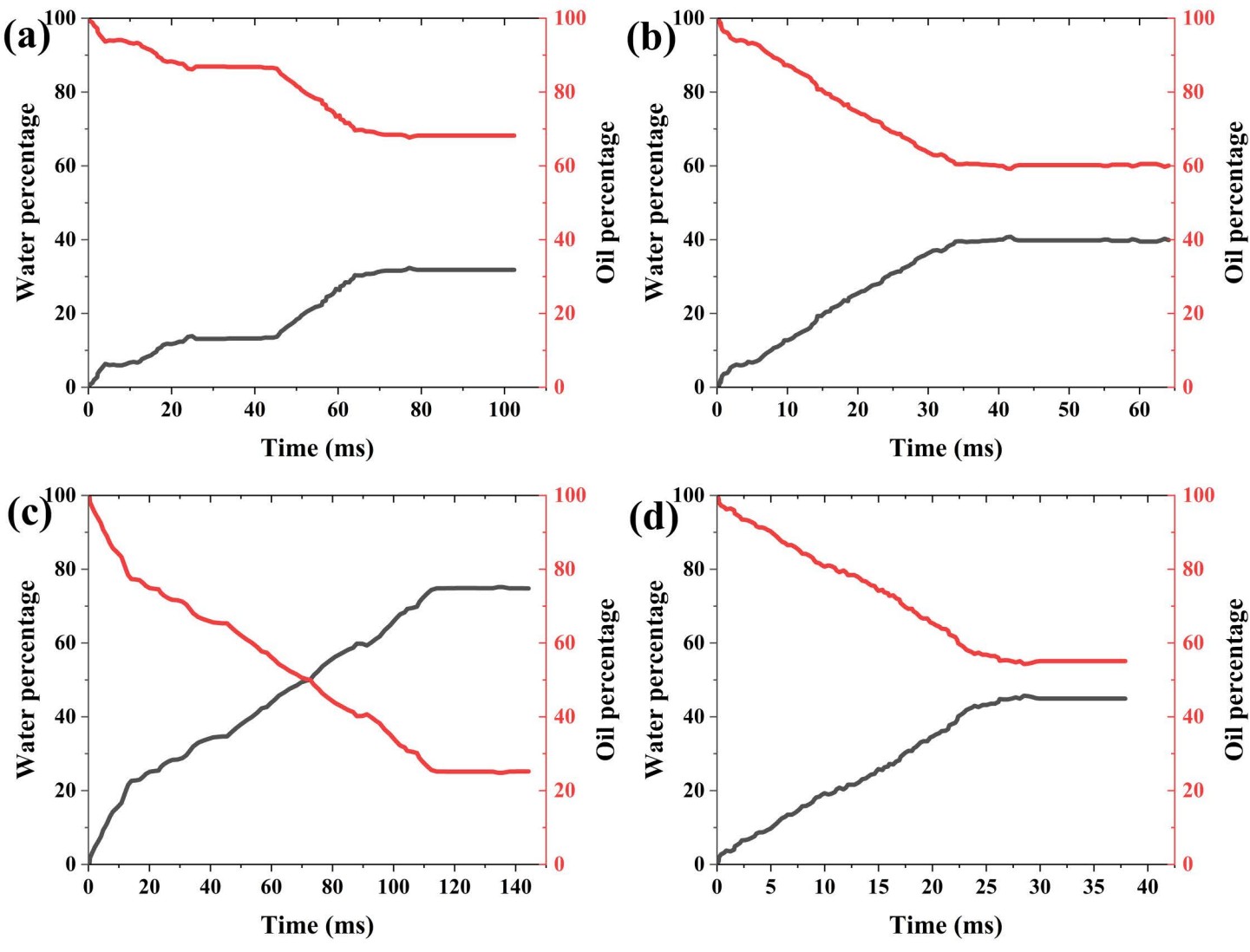

**Fig 15. Variation of oil-water mass with time and connectivity.** (a) 60% (b)70% (c) 80% (d) 90%.

interface is easy to occur in a non-uniform pore network. Here a simple analysis is processed to show the instability condition of imbibition interface.

Analysis is on one dimensional condition for simplicity (Fig 17): a throat with a length of L is initially full filled with oil. The water is imbibed into it to displace oil. The imbibition interface goes forward. Set the position of interface as ξ(t). Now let's to analysis the condition of instability.

The controlling equations for describing the water are:

$$\frac{\partial^2 p_w}{\partial x^2} = 0 \tag{9}$$

$$x = 0,\ p_w = 0,\ x = \xi,\ p_w + p_c = p_o,\ q_w = q_o \tag{10}$$

**Fig 16. Displacement efficiency and average displacement rate versus parameters.** (Longitudinal axis shows the displacement efficiency ratio means the ratio of current displacement efficiency to the displacement efficiency at baseline value. The horizontal axis shows the ratio of the currently selected parameter to the baseline value).

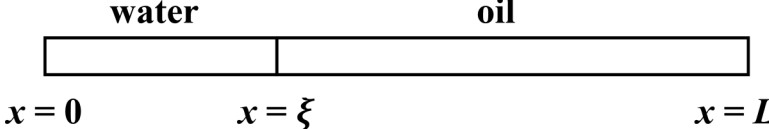

**Fig 17. Sketch of the one dimensional model.**

The controlling equations for describing the oil are:

$$\frac{\partial^2 p_o}{\partial x^2} = 0 \tag{11}$$

$$x = 0, p_o = 0, x = x, p_w + p_c = p_o, q_w = q_o \tag{12}$$

The solutions can be obtained as folows:

$$p_w = ax \ (x \leq \xi) \tag{13}$$

$$p_o = cx - cL \ (x \geq \xi) \tag{14}$$

Instituting these two equations into the condition $p_w + p_c = p_o$ and $q_w = q_o x = \xi$:

$$a\xi + p_c = c\xi - cL \tag{15}$$

$$-\frac{KK_{rw}}{\mu_w}\frac{\partial p_w}{\partial x} = -\frac{KK_{ro}}{\mu_o}\frac{\partial p_o}{\partial x} \tag{16}$$

So

$$\frac{K_{rw}}{\mu_w}a = \frac{K_{ro}}{\mu_o}c \tag{17}$$

Coefficients a, c and d can be obtained as:

$$c = \frac{Mp_c}{(M-1)\xi + L} \quad a = \frac{p_c}{(M-1)\xi - ML} \quad d = -\frac{Mp_cL}{(M-1)\xi + L} \tag{18}$$

in which $M = \dfrac{\dfrac{K_{rw}}{\mu_w}}{\dfrac{K_{ro}}{\mu_o}}$, named viscosity ratio. The velocity of imbibition interface can be expressed as:

$$\frac{d\xi}{dt} = v = q = -\frac{KK_{rw}}{\mu_w}\frac{\partial p_w}{\partial x} = \frac{KK_{rw}}{\mu_w}\frac{p_c}{(M-1)\xi + ML} \tag{19}$$

If a perturbation ε is acted on the intersurface and ε<<ξ:

$$\frac{d(\xi + \varepsilon)}{dt} = v = q = -\frac{KK_{rw}}{\mu_w}\frac{\partial p_w}{\partial x} = \frac{KK_{rw}}{\mu_w}\frac{p_c}{(M-1)(\xi + \varepsilon) + ML} \tag{20}$$

Then the evolution of the perturbation with time can be obtained as:

$$\frac{d\varepsilon}{dt} = \frac{KK_{rw}}{\mu_w}\frac{Mp_c}{(M-1)(\xi + \varepsilon) + L} - \frac{KK_{rw}}{\mu_w}\frac{p_c}{(M-1)\xi + L} \tag{21}$$

$$\frac{d\varepsilon}{dt} = \frac{Mp_cKK_{rw}}{\mu_w}\left(\frac{(M-1)\varepsilon}{[(M-1)(\xi + \varepsilon) + ML][(M-1)\xi + ML]}\right) \tag{22}$$

$$\frac{d\varepsilon}{dt} = \frac{Mp_cKK_{rw}}{\mu_w}\left(\frac{(M-1)\varepsilon}{[(M-1)\xi + L]^2}\right) \tag{23}$$

Integral of last equation gives the expression of ε with time:

$$\varepsilon = e^{\frac{Mp_cKK_{rw}}{\mu_w}\left(\frac{(M-1)}{[(M-1)\xi + L]^2}\right)t} \tag{24}$$

It is shown that the perturbation ε will increases exponentially with time t if the media is water-wetting and M > 1. That means, instability of interface or fingering will happen in this case. Since the distribution of pore size is non-uniform, the instability can happen and stop in different positions of the pore network, which leads to the non-uniform interface.

## 5. Conclusions

This paper simulates the influence of four main parameters on the spontaneous imbibition and displacement process based on the pores' distribution of a real shale sample. The results indicate that these four dimensionless factors have a significant impact on oil displacement through imbibition. From the perspective of parameter sensitivity, The connectivity has the greatest impact on average imbibition velocity and the smallest impact is the contact angle; For oil displacement efficiency, the number of capillaries has the greatest impact. The impact of main factors on imbibition and displacement is not monotonic, but rather a combination of these factors.

## Author contributions

**Conceptualization:** Zhiwei Hao.

**Funding acquisition:** Xuhui Zhang.

**Supervision:** Xiaobing Lu.

**Writing – original draft:** Tianju Wang, Li Lu.

**Writing – review & editing:** Hongsheng Guo, Yan Zhang.

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
