## [Decision Letter · Decision Letter 0]

4 Jun 2025

PONE-D-25-23798Influence of main parameters on the displacement process by spontaneous imbibition based on LBMPLOS ONE

Dear Dr. Lu,

Thank you for submitting your manuscript to PLOS ONE. After careful consideration, we feel that it has merit but does not fully meet PLOS ONE’s publication criteria as it currently stands. Therefore, we invite you to submit a revised version of the manuscript that addresses the points raised during the review process.

We look forward to receiving your revised manuscript.

Kind regards,

Hu Li

Academic Editor

PLOS ONE

“This work was supported by the National Natural Science Foundation of China (U2344223, 12302516,11872365)�the CNPC New Energy Key Project (2021DJ4902); and the High-level Innovation Research Institute Program of Guangdong Province(No.2020B0909010003).”

Reviewers' comments:

Reviewer's Responses to Questions

**Comments to the Author**

1. Is the manuscript technically sound, and do the data support the conclusions?

Reviewer #1: Yes

Reviewer #2: Yes

2. Has the statistical analysis been performed appropriately and rigorously? 

Reviewer #1: Yes

Reviewer #2: Yes

3. Have the authors made all data underlying the findings in their manuscript fully available?

Reviewer #1: Yes

Reviewer #2: Yes

4. Is the manuscript presented in an intelligible fashion and written in standard English?

Reviewer #1: Yes

Reviewer #2: Yes

5. Review Comments to the Author

Reviewer #1: The authors explore the dynamics of imbibition in shale using a color gradient model within the LBM framework combined with a realistic core pore size distribution. This manuscript contributes meaningfully to the field of imbibition in shale by leveraging a novel approach that combines LBM and dimensional analysis. Focusing on in-depth interpretation of the non-monotonic behavior in the studied parameters with respect to the base scenario, this paper has the potential to offer valuable insights for researchers and practitioners interested in multiphase flow in heterogeneous media.

However, there are some questions need to be clarified.

1 I recommend that the authors revise the text to enhance its clarity, addressing punctuation and phrasing issues.

2 In Section 2.1, it'd better to supply a figure to show the D2Q9 model. Furthermore, it is unclear in this section which parameter represents each phase; does the parameter “I” refer to one of them?

3 I suggest the authors to provide some references in recent years in the introduction.

4 During introducing the previous studies, I suggest the authors to supply some analysis of the advantages and disadvantages and limitations of others' studies, which can highlight the characteristics and advantages of this study.

Reviewer #2: I would like to express my gratitude to the editorial team for inviting me to review this manuscript. This manuscript focuses on imbibition front and phase distribution in shale based on LBM method. The topic is very interesting and the structure of manuscript is well organized. The manuscript is acceptable. Some suggestions are provided as following, which may helpful to improve the manuscript.

1. The current format of some figures is suboptimal. The authors need to improve the presentation to enhance clarity and comprehension.

2.Some typo and minor errors need to be revised to improve the English writing.

3. Suggest the authors to analyze further the mechanism of the phenomenon. and to provide some discussions on the research results.

4. The research conducted in the manuscript is intriguing; future studies could consider investigating the influence of additional pore size distribution characteristics on the permeation process.

6. PLOS authors have the option to publish the peer review history of their article (what does this mean? ). If published, this will include your full peer review and any attached files.

**Do you want your identity to be public for this peer review?** For information about this choice, including consent withdrawal, please see our Privacy Policy .

Reviewer #1: No

Reviewer #2: No

---

## [Author Response · Author response to Decision Letter 1]

13 Jun 2025

Dear Editors and reviewers:

Thank you very much for all of your comments concerning our manuscript entitled

“Influence of main parameters on the displacement process by spontaneous imbibition based on LBM). Those comments are all valuable and very helpful for improving our manuscript. We have concerned the comments carefully and made responses and revisions point by point.

[Response]: The manuscript has been restructured according to PLOS ONE's style requirements.

[Response]: LBM(color model) using in this manuscript is a widely used method in multiple fluids flow. No permits are required.

[Response]: We have supplied the related statement of "Data available statement" after section Conclusions:" The code can be available from the corresponding author upon reasonable request."

“This work was supported by the National Natural Science Foundation of China (U2344223, 12302516,11872365)�the CNPC New Energy Key Project (2021DJ4902); and the High-level Innovation Research Institute Program of Guangdong Province(No.2020B0909010003).”

[Response]: We have supplied the statement in "Funding": " The funders had no role in study design, data collection and analysis, decision to publish, or preparation of the manuscript."

[Response]: We have supplied the related statement of "Data available statement" after section Conclusions:"

[Response]: We have checked the reference list and corrected the wrong ones and added some references according to the reviewer's suggestion.

5. Review Comments to the Author

Reviewer #1: The authors explore the dynamics of imbibition in shale using a color gradient model within the LBM framework combined with a realistic core pore size distribution. This manuscript contributes meaningfully to the field of imbibition in shale by leveraging a novel approach that combines LBM and dimensional analysis. Focusing on in-depth interpretation of the non-monotonic behavior in the studied parameters with respect to the base scenario, this paper has the potential to offer valuable insights for researchers and practitioners interested in multiphase flow in heterogeneous media.

However, there are some questions need to be clarified.

1 I recommend that the authors revise the text to enhance its clarity, addressing punctuation and phrasing issues.

[Response]: Many thanks for the suggestion.

We have checked the manuscript carefully and revised the English expression. The revised contents are shown in red in the text.

2 In Section 2.1, it'd better to supply a figure to show the D2Q9 model. Furthermore, it is unclear in this section which parameter represents each phase; does the parameter “I” refer to one of them?

[Response]: Many thanks for the reviewer's suggestion.

We have provided a figure of D2Q9 model (See Figure 1) and explain the meaning of phases. "i" represents either "r" or "b", the two fluids, "r" indicates red fluid, "b" indicates blue fluid. These content are supplied in the second paragraph in section 2.2.

Fig.1 Model of D2Q9 ( 0~8 indicate the nine node numbers)

3 I suggest the authors to provide some references in recent years in the introduction.

[Response]: Many thanks for the reviewer's suggestion.

We have supplied some related research papers published in recent years.(Ref. 13-18,26)

13 Wang W, Xie Q, Wang H, Su Y, Rezaei-Gomari S. Pseudopotential-based multiple-relaxation-time lattice Boltzmann model for multicomponent and multiphase slip flow. Advances in Geo-Energy Research, 2023, 9(2): 106-116.

14 Zhang Q, Yang Y, Wang D, Sun H, Zhong J, Yao J, Lisitsa V. Correlations of residual oil distribution with pore structure during the water flooding process in sandstone reservoirs. Advances in Geo-Energy Research, 2024, 12(2): 113-126.

15 Zhou Y, Guan W, Zhao C, Zou X, He Z, Zhao H. Numerical methods to simulate spontaneous imbibition in microscopic pore structures: A review. Capillarity, 2024, 11(1): 1-21.

16 Wang H, Cai J, Su Y, Jin Z, Wang W, Li G. Imbibition behaviors in shale nanoporous media from pore-scale perspectives. Capillarity, 2023, 9(2): 32-44.

17 Cai J, Jin T, Kou J, Zou S, Xiao J, Meng Q, Lucas–Washburn Equation-Based Modeling of Capillary-Driven Flow in Porous Systems. Langmuir 2021, 37 (5): 1623-1636.

18 Cai J, Qin X, Wang H, Xia Y, Zou S. Pore-scale investigation of forced imbibition in porous rocks through interface curvature and pore topology analysis, Journal of Rock Mechanics and Geotechnical Engineering, 2025, 17(1): 245-257.

26 Lu L, Huang YD, Liu K, Zhang XH, Lu XB. Imbibition front and phase distribution in shale based on Lattice Boltzmann Method. Computer Modeling in Engineering & Sciences, 2025,142(2): 2173-2190.

4 During introducing the previous studies, I suggest the authors to supply some analysis of the advantages and disadvantages and limitations of others' studies, which can highlight the characteristics and advantages of this study.

[Response]: Many thanks for this suggestion.

We have provided some references and reviews on other's studies. Most of them are shown in section 1. For example:

Wang et al.[13] presented a pseudopotential-based multiple-relaxation-time LBM. This model can study the multicomponent flows with different molecular weights, different viscosities and different Schmidt numbers. Zhang et al.[14] studied the pore scale dynamics of imbibition in heterogeneous sandstone samples using LBM. They have not studied the dynamics of imbibition in other types of rocks. Zhou et al.[15] summarized the numerical simulation methods and research progress of spontaneous imbibition at the micro pore scale, and compared the advantages and disadvantages of various methods. Wang et al.[16] studied the imbibition in nanoporous media using LBM. Cai et al.[17] discussed the advantages and disadvantages of the classic Lucas Washburn equation, as well as its development and applications. Cai et al.[18] studied the interface dynamics and fluid-fluid interactions during imbibition of porous rocks by introducing pore topology measurement. They found the reason of unstable inlet pressure, mass flow rate, and interface curvature.

Reviewer #2: I would like to express my gratitude to the editorial team for inviting me to review this manuscript. This manuscript focuses on imbibition front and phase distribution in shale based on LBM method. The topic is very interesting and the structure of manuscript is well organized. The manuscript is acceptable. Some suggestions are provided as following, which may helpful to improve the manuscript.

1. The current format of some figures is suboptimal. The authors need to improve the presentation to enhance clarity and comprehension.

[Response]: Many thanks for this suggestion.

We have checked all the figures and supplied Figure 1, revised Figure 2, Figure 7 and Figure 10.

Fig. 2 Numerical model of baseline model

Fig.7 Relation between displacement efficiency and Ca

(The fitted curve: �=11.22+0.038e5Ca)

Fig.10 Changes of displacement efficiency of oil with contact angle. The fitted curve is �=9.38+10-3e11

2.Some typo and minor errors need to be revised to improve the English writing.

[Response]: Many thanks for the suggestion.

We have checked the manuscript carefully and revised the English expression. The revised contents are shown in red in the text.

3. Suggest the authors to analyze further the mechanism of the phenomenon. and to provide some discussions on the research results.

[Response]: Many thanks for this suggestion.

We have further analyzed the results and the mechanism of the phenomenon.(Seen in section 4.4)

4.4 Discussions

From the above results we can see that the development in the small throats is faster than that in the large throats. The path of displacement is strongly affected by the interaction among the throats (i.e., the capillary forces in these throats). According to L-W equation, the imbibition velocity in large throats is larger than that in small throats. However, the numerical results show that the imbibition velocity in small throats are larger because of the larger capillary forces which induce the displacement develops from the small throats to large throats. Therefore, in a pore-fracture network, the interaction among throats/fractures must be considered and the development of displacement in the throats with different radii must be variable. That means, the imbibition in the throats with small radius and small volume will first finish. The apparent imbibition velocity and imbibition mass will decrease gradually, which is coincidence with the experimental results[24].

Because of the non-uniform distribution of pore size, the imbibition interface does not develop smooth. In some cases, channeling path forms and so causes the decrease of displacement efficiency. The reason is that instability of interface is easy to occur in a non-uniform pore network. Here a simple analysis is processed to show the instability condition of imbibition interface.

Analysis is on one dimensional condition for simplicity (Fig. 6): a throat with a length of L is initially full filled with oil. The water is imbibed into it to displace oil. The imbibition interface goes forward. Set the position of interface as �(t). Now let's to analysis the condition of instability.

x=0 water � oil L

Fig. 6 Sketch of the one dimensional model

The controlling equations for describing the water are:

(∂^2 p_w)/(∂x^2 )=0 (9)

x=0, p_w=0; x=ξ, p_w+p_c=p_o, q_w=q_o (10)

The controlling equations for describing the oil are:

(∂^2 p_o)/(∂x^2 )=0 (11)

x=0, p_o=0; x=x,p_w+p_c=p_o, q_w=q_o (12)

The solutions can be obtained as folows:

p_w=ax (x≤ξ) (13)

p_o=cx-cL (x≥ξ) (14)

Instituting these two equations into the condition pw+pc=po and q_w=q_o at x=ξ:

aξ+p_c=cξ-cL (15)

-(KK_rw)/μ_w (∂p_w)/∂x=-(KK_ro)/μ_o (∂p_o)/∂x (16)

So

K_rw/μ_w a= K_ro/μ_o c (17)

Coefficients a, c and d can be obtained as:

c= Mp_c/((M-1)ξ+L), a= p_c/((M-1)ξ-ML) , d=-(Mp_c L)/((M-1)ξ+L) (18)

in which M= (K_rw/μ_w )⁄(K_ro/μ_o ) , named viscosity ratio. The velocity of imbibition interface can be expressed as:

dξ/dt=v=q=-(KK_rw)/μ_w (∂p_w)/∂x=(KK_rw)/μ_w p_c/((M-1)ξ+ML) (19)

If a perturbation � is acted on the intersurface and �<<ξ:

d(ξ+ε)/dt=v=q=-(KK_rw)/μ_w (∂p_w)/∂x=(KK_rw)/μ_w p_c/((M-1)(ξ+ε)+ML) (20)

Then the evolution of the perturbation with time can be obtained as:

dε/dt=(KK_rw)/μ_w (Mp_c)/((M-1)(ξ+ε)+L)-(KK_rw)/μ_w p_c/((M-1)ξ+L) (21)

dε/dt=(Mp_c KK_rw)/μ_w ((M-1)ε/[(M-1)(ξ+ε)+ML][(M-1)ξ+ML] ) (22)

dε/dt=(Mp_c KK_rw)/μ_w ((M-1)ε/[(M-1)ξ+L]^2 ) (23)

Integral of last equation gives the expression of ε with time:

ε=e^((Mp_c KK_rw)/μ_w (((M-1))/[(M-1)ξ+L]^2 )t) (24)

It is shown that the perturbation ε will increases exponentially with time t if the media is water-wetting and M>1. That means, instability of interface or fingering will happen in this case. Since the distribution of pore size is non-uniform, the instability can happen and stop in different positions of the pore network, which leads to the non-uniform interface.

4. The research conducted in the manuscript is intriguing; future studies could consider investigating the influence of additional pore size distribution characteristics on the permeation process.

[Response]: Many thanks for this suggestion.

In the future studies, we will investigate the influence of pore size distribution characteristics on the permeation process to make the results more practicable.

---

## [Editor Report · Decision Letter 1]

4 Jul 2025

Influence of main parameters on the displacement process by spontaneous imbibition based on LBM

PONE-D-25-23798R1

Dear Dr. Lu,

We’re pleased to inform you that your manuscript has been judged scientifically suitable for publication and will be formally accepted for publication once it meets all outstanding technical requirements.

Kind regards,

Hu Li

Academic Editor

PLOS ONE
---

## [Editor Report · Acceptance letter]

PONE-D-25-23798R1

PLOS ONE

Dear Dr. Lu,

I'm pleased to inform you that your manuscript has been deemed suitable for publication in PLOS ONE. Congratulations! Your manuscript is now being handed over to our production team.

Kind regards,

on behalf of

Pro.Dr. Hu Li

Academic Editor

PLOS ONE